# Analytical Analysis of Factors Affecting the Accuracy of a Dual-Heat Flux Core Body Temperature Sensor

**DOI:** 10.3390/s24061887

**Published:** 2024-03-15

**Authors:** Jakub Żmigrodzki, Szymon Cygan, Jan Łusakowski, Patryk Lamprecht

**Affiliations:** 1Institute of Metrology and Biomedical Engineering, Faculty of Mechatronics, Warsaw University of Technology, 02-525 Warsaw, Poland; szymon.cygan@pw.edu.pl (S.C.); jan.lusakowski.stud@pw.edu.pl (J.Ł.); 2Healthwear Sp. z o.o., 31-539 Cracow, Poland; plamprecht@inviswearables.com

**Keywords:** core body temperature, dual heat-flux sensor, Monte Carlo, measurement accuracy

## Abstract

Non-invasive core body temperature (CBT) measurements using temperature and heat-flux have become popular in health, sports, work safety, and general well-being applications. This research aimed to evaluate two commonly used sensor designs: those that combine heat flux and temperature sensors, and those with four temperature sensors. We used analytical methods, particularly uncertainty analysis calculus and Monte Carlo simulations, to analyse measurement accuracy, which depends on the accuracy of the temperature and flux sensors, mechanical construction parameters (such as heat transfer coefficient), ambient air temperature, and CBT values. The results show the relationship between the accuracy of each measurement method variant and various sensor parameters, indicating their suitability for different scenarios. All measurement variants showed unstable behaviour around the point where ambient temperature equals CBT. The ratio of the heat transfer coefficients of the dual-heat flux (DHF) sensor’s channels impacts the CBT estimation uncertainty. An analysis of the individual components of uncertainty in CBT estimates reveals that the accuracy of temperature sensors significantly impacts the overall uncertainty of the CBT measurement. We also calculated the theoretical limits of measurement uncertainty, which varied depending on the method variant and could be as low as 0.05 °C.

## 1. Introduction

Non-invasive continuous measurement of core body temperature (CBT) is used in a variety of applications, including control of body stress caused by exercise and weather, medical diagnostics, or assessment of thermal comfort.

The first area includes applications related to health protection against dangerous work conditions and work intensity. The significant burden placed on the body by effort and thermal conditions can be dangerous to health and life. Quantitatively measuring and rating this load is essential for people who work physically in hard conditions (such as high humidity and temperature), for example firefighters, soldiers, miners, and professional athletes. The quantitative assessment of thermal and physical exertion enables an objective rating of workload, while also reducing the risk of life-threatening situations. CBT increase is one of the symptoms of a heat-related illness [1]. Physiological load is related to many parameters, such as surface body temperature (SBT) and CBT, intensity of sweating, and heart rate. In the last few decades, many quantitative methods of assessing physiological load have been proposed. Among these methods, one should distinguish the Physiological Strain Index (PSI) proposed by Moran in 1998, which, despite a simple formula based only on CBT and heart rate, has shown its usefulness [2,3]. The method’s sensitivity to weather conditions, clothing (e.g., personal protective equipment), dehydration, and sex has been demonstrated. An update to the PSI to also consider skin temperature, proposed by Buller in 2016, showed strong sensitivity to clothing [4]. Despite all its advantages, a major problem in the common use of the PSI is the difficulty in measuring CBT while ensuring comfort and without interfering with the subject’s daily life. In sports medicine, the measurement of CBT is essential for assessing the risk of exercise-induced heat stroke, which can lead to an athlete’s death [5]. It is estimated that 1 in 1000 athletes exercising at high ambient temperatures will experience exercise-induced heat stroke. It is assumed that athletes whose internal body temperature exceeds 40 °C, and who exhibit symptoms of central nervous system disorders, should receive immediate medical care.

The second area includes medical applications. Continuous monitoring of CBT is not only useful in detecting and monitoring fever in infectious diseases but is also an important parameter in the evaluation of severe medical conditions such as sepsis [6]. The monitoring of daily temperature fluctuations has proved useful in assessing diurnal rhythm disturbances, which may cause mood swings, appetite disorders, affect work performance, and can be related to delayed phase syndrome, or be a symptom of mental or neurological conditions such as depression, bipolar affective disorder, or schizophrenia [7,8,9,10,11]. CBT may be useful in evaluating diets [12], assessing the menstrual cycle, predicting fertile days and timing of pregnancy, and detecting potentially threatened preterm labour [13,14]. It should be noted that the medical community has expressed a demand for new, more accurate solutions and standards in this area [15].

A literature review showed the existence of methods for contact CBT measurement using skin surface temperature sensors and heat flux through the surface of the body, which might be a solution for increasing the comfort of the measurement which is currently the main limitation to the wider application of the PSI [4,16,17,18,19,20]. The available results have shown that CBT measurement using such methods can provide accuracy as high as 0.09 ± 0.05 °C in laboratory conditions [21]. Accuracy achieved in clinical conditions is approximately ±0.3 °C [20,22]. The available data suggest that CBT can be measured using wrist-based sensors with an accuracy of <0.2 °C [23]. Importantly, several CBT sensors are commercially available, including Tcore (Dräger Medical GmbH, Lübeck, Germany), CORE (Greenteg, Rümlang, Switzerland), ScanWatch 2 (Withings, Issy-les-Moulineaux, France), Temple Touch Pro™ (Medisim, Beit-Shemesh, Israel), and BairHugger™ (3M™, Maplewood, MN, USA), of which the first three are wearable devices and last two are systems designed for hospitals.

The method for CBT measurement using a non-invasive sensor compromising the temperature and/or heat flux sensor is a relatively new area of scientific research, which is gaining the interest of research teams around the word. One direction is research of construction or technological modification, for example by developing flexible sensors [24,25], reducing sensitivity to convective conditions [26], or adapting sensors to the system requirements of the Internet of Things (IoT) [27]. Another fast-growing research area is the usability of such technology in clinical [28,29], sport [30], work safety [27] or free-living [26,31] applications. It is worth noting that new patents are also being sought, which confirms the commercial sector’s interest in this technology [32].

Measurement uncertainty is a crucial aspect of every measurement method. Therefore, the authors deemed it essential to investigate this aspect of dual-heat flux CBT estimation methods while also proposing some new modifications to the measurement methods. The authors are aware of only a single publication that analyses the factors impacting the measurement uncertainty of skin-contact thermal resistance [33]. Additionally, another publication examines the influence of sensor structure on measurement accuracy [22]. The aim of the work conducted and presented herein was to analyse the selected factors that influence the accuracy of the non-invasive CBT measurement using dual-heat flux sensors. Both commonly used CBT sensor designs were included in the research: one with two heat flux sensors and two temperature sensors and one with four temperature sensors. The factors analysed were the accuracy of temperature and flux sensors, as well as the mechanical construction parameters of sensors like the heat transfer coefficient, temperature of ambient air, and CBT value. The influence of these factors was assessed using analytical methods such as uncertainty analysis calculus and Monte Carlo simulations. The results of the experiments offer a valuable source of knowledge for engineers and scientists working on non-invasive CBT measurement methods.

## 2. Materials and Methods

A simplified model of the human body’s heat generation and distribution system was used, and CBT was assumed to be equal to the temperature of the blood perfusing through the hypodermis. The skin was considered as a thermally passive layer between blood and the environment (air). All layers (blood, skin, and air) were assumed to be infinite in directions orthogonal to tissue cross-section and the temperature of blood was constant in the whole layer. In such a system, CBT estimate (CBT_e_) can be expressed in terms of heat flux q flowing through the sensor, sensor heat transfer coefficient h_s_, and skin surface temperature T_ss_, as [17,20,21,34,35,36]:(1)$$CBT_{e}=T_{ss}+\frac{q}{h_{t}}$$
where CBT_e_—body core temperature estimate; T_ss_—temperature of skin surface; q—heat flux flowing through skin and sensor; and h_t_—heat transfer coefficient of the skin.

The heat flux q can be measured directly by the heat flux sensor placed between the skin and sensor material, or indirectly using two temperature sensors measuring the temperature difference between both sides of the sensor’s thermally resistive layer. It can be expressed as:(2)$$q=h_{s}\cdot (T_{ss}-T_{sa})$$
where q—heat flux flowing through skin and sensor; h_s_—heat transfer coefficient of the thermally resistive layer of the sensor; and T_ss_ and T_sa_—temperature of the bottom (skin–sensor) and top (sensor–air) surfaces.

For clarity, the first variant of the single-flux method comprising a heat flux sensor and a temperature sensor will be denoted by the abbreviation “HT”, and the second variant comprising two temperature sensors will be denoted as “TT”. 

Even though there are commercially available products using the method described above (single-flux method), like Tcore (Dräger Medical GmbH, Germany), the practical application of single-sensor CBT measurement devices is limited due to the dependence of the CBT estimate on the thermal properties of the skin (h_t_). As h_t_ is a function of tissue thickness and thermal conductivity, the application of a single sensor is limited to the places in which variation of this parameter in population is narrow, e.g., the forehead. To overcome this limitation, the dual-heat flux method was introduced [17,21,35,37]. In this approach, two sensors positioned near each other are used (Figure 1).

If two heat flux sensors are used, two independent equations for CBT_e_ can be formulated.
(3)$$CBT_{e}=T_{ss1}+\frac{q_{1}}{h_{t}}$$
(4)$$CBT_{e}=T_{ss2}+\frac{q_{2}}{h_{t}}$$

Using Equations (3) and (4), the formula for h_t_ can be derived.
(5)$$h_{t}=\frac{q_{1}-q_{2}}{T_{ss1}-T_{ss2}}$$

Finally, the formulae for a CBT estimate independent of tissue properties can be obtained by substituting Equation (5) in Equation (3). If the HT configuration is used, the resulting equation is:(6)$$CBT_{eHT}=T_{ss1}+\frac{q_{1}\cdot (T_{ss1}-T_{ss2})}{q_{2}-q_{1}}$$

If the TT configuration is used, Equation (6) can be expressed as:(7)$$CBT_{eTT}=T_{ss1}+\frac{\left(T_{ss1}-T_{sa1})(T_{ss1}-T_{ss2}\right)}{k\cdot (T_{ss2}-T_{sa2})-(T_{ss1}-T_{sa1})}$$
where k is the characteristic parameter of the sensor and equals h_s2_/h_s1_.

One can notice that Equations (6) and (7) make use of the difference between temperature and heat flux measured in both channels. The error of this differential measure can be reduced by sensor pairing and estimating constant bias between both sensors. In the case of temperature sensors, this can be easily achieved by submerging both sensors in the same fluid, i.e., an oil bath, and measuring the difference between temperature readings from both sensors. Then, the correction factor can be introduced. If the initial bias between sensors is corrected, the only source of uncertainty of temperature difference is their internal noise. Theoretically, this can also be undertaken to reduce the error of heat flux difference measurement, but the process is more complex. For simplicity, in this work, only the influence of temperature difference calibration will be considered. As the uncertainty of measurement of the temperature difference will be different from absolute temperature measurement, the approach with a pair of calibrated sensors will be treated as another variant, for which Equations (6) and (7) are expressed as follows:(8)$$CBT_{eHTp}=T_{ss1}+\frac{q_{1}\cdot \Delta T_{ss}}{\left(q_{2}-q_{1}\right)}$$
(9)$$CBT_{eHTp}=T_{ss1}+\frac{\Delta T_{s1}\cdot \Delta T_{ss}}{k\cdot \Delta T_{s2}-\Delta T_{s1}}$$
where:(10)$$\Delta T_{ss}=T_{ss1}-T_{ss2}$$
(11)$$\Delta T_{s1}=T_{ss1}-T_{sa1}$$
(12)$$\Delta T_{s2}=T_{ss2}-T_{sa2}$$

Equations (8) and (9) can be transformed as a function of the readings from temperature sensors other than T_ss1_. Equation (8) can be expressed as:(13)$$CBT_{eHTp}=T_{ss2}+\frac{q_{2}\cdot \Delta T_{ss}}{\left(q_{2}-q_{1}\right)}$$
and Equation (9) can be transformed into three other forms:(14)$$CBT_{eTTp}=T_{ss2}+\Delta T_{ss}+\frac{\Delta T_{s1}\cdot \Delta T_{ss}}{k\cdot \Delta T_{s2}-\Delta T_{s1}}$$
(15)$$CBT_{eTTp}=T_{sa1}+\Delta T_{s1}+\frac{\Delta T_{s1}\cdot \Delta T_{ss}}{k\cdot \Delta T_{s2}-\Delta T_{s1}}$$
(16)$$CBT_{eTTp}=T_{sa2}+\Delta T_{ss}+\Delta T_{s2}+\frac{\Delta T_{s1}\cdot \Delta T_{ss}}{k\cdot \Delta T_{s2}-\Delta T_{s1}}$$

Equations (8) and (13), as well as Equations (9), (14), (15), and (16), can be considered as independent measurements whose values can be averaged, which should reduce the CBT estimation uncertainty. The averaged results from the procedure will be termed CBT_eHTpm_ when only Equations (8) and (13) are used, and CBT_eTTpm_ when Equations (9), (14), (15), and (16) are used.

For clarity, all the used measurement strategies for which the uncertainty was validated are summed up in Table 1.

### 2.1. CBT Measurement Uncertainty Analysis

The influence of measurement setup parameters, such as ambient and CBT temperatures, values and ratios of the h_s_ parameters of both channels and values of the uncertainty of input measurement factors, e.g., heat flux and temperature, was investigated using the Monte Carlo (MC) method. This method was chosen due to the high nonlinearity of Equations (6)–(9). Next, in the linear region of Equations (6)–(9) (estimated using MC simulation), the contribution of each uncertainty source to the total CBT measurement uncertainty was studied using the linearization method based on a first-order Taylor expansion of Equations (6)–(9). This analysis was carried out for the HT, HTp, TT, and TTp measurement strategies (Table 1).

#### 2.1.1. Measurement Setup

The estimation of CBT measurement uncertainty using the dual-flux method requires a prediction of the expected value of measurement from each sensor embedded in the dual-flux probe, i.e., the temperature and flux sensors. These values provided a reference to which random error was added to simulate the uncertainty of the measurements. The reference values of temperature and heat flux readings from each sensor composed into a dual-heat flux probe were calculated using simplified lumped models presented in Figure 1. The parameters of this model are presented in Table 2.

The calculation of the q_1_, T_ss1_, and T_sa1_ in channel 1 was conducted according to the following equations:(17)$$h_{1}=\frac{1}{\frac{1}{h_{t}}+\frac{1}{h_{s1}}+\frac{1}{h_{sa}}}$$
(18)$$q_{1}=h_{1}\cdot (CBT-T_{amb})$$
(19)$$T_{ss1}=CBT-\frac{q_{1}}{h_{t}}$$
(20)$$T_{sa1}=T_{ss1}-\frac{q_{1}}{h_{s}}$$

An analogous calculation leads to values of q_2_, T_ss2_, and T_sa2_ in the second channel of the probe.

Using the above methodology, a simulation of the expected values of the q_1_, q_2_, T_ss1_, T_ss2_, T_sa1_, and T_sa2_ for different values of T_amb_ and CBT was conducted. The influence of T_amb_ and CBT values was investigated independently in the ranges of −15 °C to +45 °C and 35 °C to 42 °C, respectively.

#### 2.1.2. Monte Carlo Analysis

The main procedure of CBT uncertainty estimation involves the following steps: calculation of the expected (error-free) values of measurements (using Equations (18)–(20); calculation of the noisy values of the measurements by adding a random value to each measurement (as described below); calculation of the CBT value estimate for each measurement variant (Table 1) using Equations (6)–(16) and noisy values of the heat flux and temperature measurement; and calculation of the CBT estimation error by subtracting the CBT referential value from the CBT estimate. This procedure was repeated 10,000 times for each set of the experimental setup parameters (Table 1 and Table 2), while the value of the noise was sampled from Gaussian distribution (Table 3). The CBT estimate uncertainty was estimated for the selected measurements variant (Table 1) as the standard deviation of estimate errors from 10,000 repetitions. The CBT estimate uncertainty estimation was repeated 100 times to estimate the repeatability of MC analysis results. Finally, the mean mu_CBT_ and standard deviation su_CBT_ of the CBT estimate uncertainty derived for 100 trials were calculated. The mu_CBT_ constitutes the prediction of the standard uncertainty value of the CBT estimate using selected measurement variants and the measurement setup parameter values. The su_CBT_ is the measure of the repeatability of the MC simulation for a given measurement method and setup configuration and can be used as a measure of confidence in the estimation of the mu_CBT_. The noisy measurements were calculated by assuming that the error values originate from a normal distribution with a mean value of 0. The standard deviation for each type of error source is denoted as u_T_, u_ΔT_, u_q_, and u_h_, representing measurement errors of non-paired temperature, differential temperature using paired sensors, heat flux, and heat transfer coefficient of measurement channels, respectively. The default values for these uncertainties, along with their justifications, are presented in Table 3.

Using the MC method, the impact of serval factors on CBT estimation uncertainty was investigated. The study was divided into a series of independent experiments. In each experiment, only the value of one parameter of the measurement setup was changed while other parameters presented in Table 2 and Table 3 were set to their default values. The list of all conducted experiments is presented in Table 4.

#### 2.1.3. First-Order Tylor Series Expansion

For the dual-flux sensor in HT with non-paired temperature sensor configuration (Equation (6)), the combined uncertainty u_CBTeHT_ can be expressed using first-order Tylor series expansion as:(21)$$u_{CBTeHT}(T_{ss1},T_{ss2},q_{1},q_{2})=\sqrt{\begin{array}{c} (\frac{\delta CBT_{eHT}}{\delta T_{ss1}}{)}^{2}\cdot u_{T}^{2}+(\frac{\delta CBT_{eHT}}{\delta T_{ss2}}{)}^{2}\cdot u_{T}^{2}+\\ (\frac{\delta CBT_{eHT}}{\delta q_{1}}{)}^{2}\cdot u_{q}^{2}+(\frac{\delta CBT_{eHT}}{\delta q_{2}}{)}^{2}\cdot u_{q}^{2} \end{array}}$$
where u_T_ and u_q_—measurement uncertainty of the applied temperature and heat flux sensors, which were assumed to be identical for all heat flux and temperature sensors.

The ratio between each factor uncertainty component and total uncertainty was calculated as a measure (ru) of the share of each component in the total measurement uncertainty. The equation for this ratio for the T_ss1_ component is defined as follows:(22)$$ru(T_{ss1})=\frac{(\frac{\delta CBT_{eHT}}{\delta T_{ss1}}{)}^{2}\cdot u_{T}^{2}}{u_{CBTeHT}^{2}}$$

Analogously to this equation, ru values were calculated for the remaining factors, i.e., T_ss2_, q_1_, and q_2_.

The ru values for other measurement configurations, e.g., HTp, TT, and TTp, were calculated using the same approach. The ru coefficients for all cases were calculated following Equation (22), using the setup and uncertainty parameter values set to the default values, as presented in Table 2 and Table 3.

### 2.2. Tools

The analysis of the uncertainty of CBT estimates using the MC simulations and Taylor series expansions method was conducted using Python 3.10 and numpy, sympy, random, statistics, and matplotlib packages.

## 3. Results

### 3.1. Experimental Setup Validation

In the first experiment, the values of heat flux flowing through the sensors model and temperature spot on both sides of sensors were estimated for different values of ambient temperature and constant value of CBT, which equals 37 °C. Other parameters of the experimental setup were set to their default values (Table 2). The values of heat flux q_1_ and q_2_ are in the approximate range from −30 to 192 W/m^2^. These values are comparable with the values reported by Niedermann, R. et al. [18]—the mean value of heat flux measured at the chest equals 201 W/m^2^. As expected, when T_amb_ equals CBT, both heat fluxes are zero. The T_ss_ values are in the range from approximately 35 °C to 38 °C, as T_sa_ spans between approximately 35 °C and 40 °C.

Next, the values of heat flux flowing through the sensors model and temperatures of both sides of the sensors were estimated for different values of CBT and a constant value of T_amb_ equal to 25 °C. As expected, due to the higher value of the heat transfer coefficient of channel 1, the value q_1_ is higher than q_2_. The difference in both measured heat fluxes grows as the CBT increases from 10 W/m^2^ to 17 W/m^2^. Similar effects are observed for temperatures T_ss_ and T_sa_. The difference between T_ss2_ and T_ss1_ varies from 0.07 °C to 0.11 °C as the difference between T_sa2_ and T_sa1_ varies from 1.64 °C to 2.78 °C.

### 3.2. Monte Carlo Simulations

Figure 2, Figure 3 and Figure 4 present results of the estimation of the CBT measurement uncertainty (mu_cbt_) and a standard deviation of these estimates (su_cbt_) in the function of the ambient temperature (T_amb_). The obtained results suggest that all measurement variants show stable and unstable behaviours in different ranges of T_amb_. In the unstable range of T_amb_ values, a rapid increase in the mu_cbt_ and su_cbt_ is observed. The ratio between the maximum values of mu_cbt_ and su_cbt_ obtained in the unstable regions and the maximum values denoted for stable regions can reach values ranging from hundreds to thousands. For all methods, the unstable results are yielded around the point where T_amb_ equals CBT. The variants using the heat flux sensors (HT, HTp, and HTpm) show the widest range of unstable behaviour, starting approximately at T_amb_ of 27–28 °C. The TT variant shows moderate width of unstable behaviour reaching from 33.8 °C to 42.3 °C. The TTp and TTpm show the lowest range of T_amb_, resulting in unstable behaviour spreading from 36.98 °C to 37.02 °C. Also, different tendencies of mu_cbt_ in stable regions are observed for each measurement variant (Figure 3). The values of mu_cbt_ decrease as T_amb_ rises to the point where T_amb_ equals CBT, and then start to increase in cases where the measurement variants use only temperature probes (TT, TTp, and TTpm). The HT, HTp, and HTpm variants only show a minor increase in mu_cbt_ in the cases where T_amb_ is located in the stable behaviour range. The measurement variants using paired temperature sensors show smaller mu_cbt_ values than variants using unpaired sensors. The differences in mu_cbt_ are in the range from approx. 0.075 °C to 1.25 °C for TT type variants and 0.08 °C for HT type variants. The use of multi-measurement techniques (HTpm and TTpm) only slightly reduced mu_cbt_ as compared to the variants with paired temperature sensors.

The influence of CBT value in the range of temperature typical for the human body on mu_cbt_ and su_cbt_ is limited (Figure 5). The value of mu_cbt_ for un-paired temperature sensors is less than 0.15 °C in the whole investigated range of CBT. For other variants, mu_cbt_ is less than 0.06 °C.

The CBT estimation error (mu_cbt_) value depends on the ratio between the heat transfer coefficients of both channels (Figure 6). The mu_cbt_ value increases as the ratio between h_s1_ and h_s2_ approaches 1. The narrowest stable region is observed for the TT variant for which the mu_cbt_ starts to grow rapidly for a heat transfer coefficient ratio above approximately 24. The minimum value of mu_cbt_ for TT is observed for the h_s2_/h_s1_ ratio equal to 7. In the case of HT variants (HT, HTp and HTpm), the mu_cbt_ decreases steadily as the h_s2_/h_s1_ ratio grows in the whole investigated range. The values of mu_cbt_ start to settle as h_s2_/h_s1_ ratios reach approximately 100, 10, and 10 for HT, HTp, and HTpm variants, respectively. The minimum value of the mu_cbt_ reaches values of approx. 0.097 °C, 0.04 °C, and 0.035 °C for HT, HTp, and HTpm variants, respectively. The mu_cbt_ starts to rapidly grow as the h_s2_/h_s1_ ratio reaches approximately 2440 for TTp and TTpm alike. The minimum value of the mu_cbt_ reaches values of approx. 0.034 °C and 0.024 °C for TTp and TTpm variants, respectively. Those minima are located for an h_s2_/h_s1_ ratio equal to 30.

The optimal value of the heat transfer coefficient of channel 1 (h_s1_) varies with the measurement variant (Figure 7). The narrowest range of h_s1_ (from 1.9 W/(m^2^·K) to 50.2 W/(m^2^·K)) resulting in minor changes in the mu_cbt_ is observed for the method using unpaired temperature sensors (TT). The widest range (from 1.91 W/(m^2^·K) to 534.7 W/(m^2^·K)) is observed for TTp and TTpm variants. The lowest values of the mu_cbt_ were noticed for HTp and HTpm. The locations of the minima for these two variants are found for relatively low h_s1_ values equal to approximately 1.7 W/(m^2^·K).

The value of the h_sa_ parameter, which is a measure of convective heat transfer from sensors to the surrounding air, shows a variable influence on mu_cbt_ (Figure 8). The most significant impact is observed for HT and TT measurement variants. In these cases, the mu_cbt_ lowered from 0.14 °C to approximately 0.08 °C as the h_as_ changed from 6 W/(m^2^·K) to 104 W/(m^2^·K). For the HTp and HTpm variants, the mu_cbt_ reduction is limited to approximately 0.017 °C. The mu_cbt_ measured for TTp and TTpm variants only show marginal dependency on the h_sa_ value in the investigated range.

### 3.3. Linearization

The relative share of each type of independent uncertainty source to the total uncertainty of CBT measurement varied between studied variants of measurement (Table 5). In all cases, except the HTp variant, the direct (non-differential) temperature measurement constitutes the main source (~90%) of CBT estimate uncertainty. In the case of the HTp variant, the temperature and heat flux measurements contribute approximately equally to the overall uncertainty.

## 4. Discussion

The human body’s thermoregulation, heat generation, and transfer processes are complex, and therefore challenging to model. Even establishing a clear and unambiguous definition of CBT is not a trivial task [45]. For this reason, the analytical examination of the interaction between the body and the measuring device requires simplification. The assumptions and constraints imposed on the thermal model used for the heat generation and distribution system significantly simplified the description of the problem. The employed layer model is justified by the interaction between the sensor and the body in scenarios where typical sensor areas, such as the forehead, wrist, sternum, or side of the chest, are used for measurement. This simplification is additionally justified by the size of the measurement area, which typically is in the range of 4 cm^2^ to 15 cm^2^ [17,21,35,36,37]. 

The predicted ranges of readings from sensors embedded in the dual-heat flux CBT probe seem to correlate in size with the values found in the literature. The relation of these values to changes in ambient temperature and CBT confirms our expectations, which verifies the proposed methodology.

The results of the MC simulations show a strong influence of the ambient temperature on CBT estimate uncertainty. All variants of measurement show unstable behaviour around the point where T_amb_ equals CBT, where the heat flux in both channels equals zero and all temperature readings have the same value. This leads to an infinite number of solutions to the equation system formed by Equations (3) and (4). The range of those unstable regions is related to the CBT measurement variant. The widest unstable regions are observed for variants using direct heat flux sensors for which stable regions end at T_amb_ reaching 27–28 °C. This seems to be a significant limitation of those measurement variants, as this level of ambient temperature is common in many regions of the world. Moreover, this feature excludes these variants from applications in which monitoring of thermal stress is important, such as monitoring of firefighters, steelworks staff, or sports professionals. A narrower unstable region was observed for probes containing only paired temperature sensors (TTp and TTpm) for which measurements are unpredictable only in the span of 0.04 °C around the CBT value. The advantage of HT-type methods is the lower value of mu_cbt_ in the range of T_amb_ below 10 °C as compared to TT-type measurement variants. This can be important in applications of CBT measurement in cold environments, such as monitoring staff in cold stores or high-altitude mountaineering.

The CBT value has limited impact on its estimate uncertainty if the value of T_amb_ is far from the CBT value and thus measurement is conducted in the stable region. The results of the experiments indicate that when using paired temperature sensors, uncertainty around 0.05 °C could be achieved in the range of CBTs spanning from 35 °C to 42 °C. Such measurement accuracy seems to be acceptable in the majority of applications, including medical temperature measurement [46].

The ratio of the heat transfer coefficients h_s2_/h_s1_ has an impact on the CBT estimation uncertainty (mu_cbt_). The optimal value of this ratio is specific to the measurement variant. For all HT type variants and TTp and TTpm variants, the optimal value of the ratio is between 10 and 100. In the case of the TT variant, the optimum is reached for a ratio equal to approximately seven. For all methods, mu_cbt_ rises with the h_s2_/h_s1_ ratio approaching one. The ratio of the heat transfer coefficients equal to one represents a scenario in which both channels are identical, which leads to an infinite number of solutions to the equation system formed by Equations (3) and (4). According to the obtained results, the ratio of the heat transfer coefficients should be set in the range of 3 to 10.

The use of paired temperature sensors (TTp and TTpm) results in a wider range of h_s_ values, yielding minor changes in CBT uncertainty to its minimal value. This can be considered an additional benefit providing less restrictive design requirements for the dimensions and material of the dual-heat flux sensor. The relatively high values of h_s_ for which minimum values of mu_cbt_ are denoted in the cases of TTp and TTpm variants indicate that these variants should be considered in the first place when significant miniaturization of dual-heat flux probes is planned. The reduction in the thickness of measurement channels can increase h_s_ beyond the stable range of other variants. Meanwhile, HTp and HTpm variants provide the lowest mu_cbt_ values which can be beneficial in the cases when the size of the probe is not a major design limitation.

The investigated variants of measurements show varied sensitivity to the intensity of heat transfer between the sensor and the surroundings. This factor becomes crucial in the design of a dual-heat flux CBT probe intended for operation in various insulated environments, such as underneath clothing or protective suits. In such applications, h_sa_ is expected to be relatively low and the use of TTp or TTpm approaches could be considered beneficial due to their low estimation error. 

The uncertainty sources analysis shows that in all measurement variants, the application of more precise temperature sensors can lead to a significant reduction in total CBT estimation uncertainty. The designers should therefore mainly focus on searching for temperature sensors with accuracy higher than 0.05 °C to further improve the accuracy of dual-heat flux sensors.

## 5. Conclusions

The presented analysis compares six different variants of dual-heat flux measurement of CBT. The expected values of the uncertainty of CBT estimates obtained using these variants were estimated. The sensitivity of measurement uncertainty to factors of measurement setup such as ambient temperature, CBT, the values and ratios of the heat transfer coefficients of the dual-heat flux probe channels, and the intensity of the convective heat dissipation rate was investigated using the Monte Carlo method. 

The most important observations resulting from the analysis are as follows: In the ranges of T_amb_ close to CBT, the measurement uncertainty is significantly high. This phenomenon limits the applications of the analysed CBT measurement methods when the ambient temperatures are close to CBT, i.e., in hot climates or work environments. These limitations are different for each measurement variant and some, such asTTp and TTpm, could be considered as not definitively excluding it from application in conditions of increased ambient temperature, as the unstable range is insufficiently broad for these variants. For all studied variants, the expected value of measurement uncertainty is less than 0.1 °C in the limited range of ambient temperatures below CBT. This indicates the usefulness of this measurement technique in medical applications, as the standards for electronic thermometers allow for an acceptable error reaching 0.1 °C [46]. 

For some measurement variants and in specific configurations, the predicted standard uncertainty of measurement can be lower than 0.05 °C. These results should be treated as a theoretical lower boundary of uncertainty, estimated using only analytical methods and not confirmed by empirical experiments. Another important observation is the specific range of optimal values of heat transfer coefficients of the dual-heat flux probe channels. This indicates that the material and mechanical design of the sensor should be adapted to the specific variant of measurement.

The TTp and TTpm variants allow for the use of relatively high values of the h_s_ at which the minimum value of uncertainty is denoted. This feature predestines these variants for applications in miniature sensors, as obtaining low values of heat transfer coefficients in such cases can be a challenge. 

An analysis of the individual components of uncertainty in CBT estimates reveals that, when employing temperature and heat flux sensors available on the market, the accuracy of temperature sensors has the most significant impact on the overall uncertainty of the CBT measurement. Therefore, it represents the primary limitation of the method. The conducted research showed that using paired temperature sensors could substantially decrease the uncertainty of the CBT measurement.

The main limitation of the presented study is the lack of empirical validation and the use of a simplified model. The former should be an inspiration for the continuation of this research. The latter limitation could be justified by the preliminary form of the study, as well as the need for simplification of the very complex system of the human body and its thermal interaction with the environment. Without such simplification, the number of parameters which should be included in the analysis would reduce the clarity and interpretability of the results. The values of the predicted readings from the sensors using the proposed model correspond with the values measured in vivo, which are publicly available. This seems to confirm the usefulness of the applied models and methods. The obtained values of uncertainty correlate with results obtained in empirical tests conducted by other authors (see Table 6), further supporting the presented methodology.

## Figures and Tables

**Figure 1 sensors-24-01887-f001:**
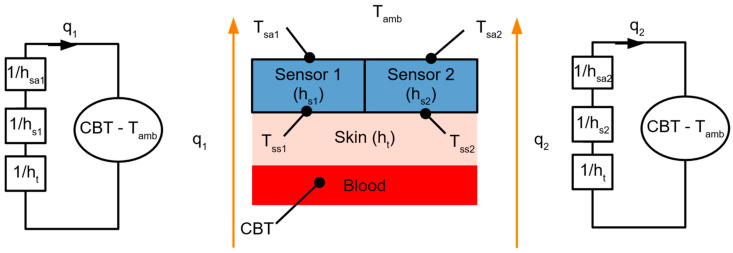
The CBT measurement setup for a dual flux sensor and its looped model equivalent. T_amb_—ambient (air) temperature, T_sa1_ and T_sa2_—temperature measured between air and sensors 1 and 2, respectively, T_ss1_ and T_ss2_—temperature measured between skin and sensors 1 and 2, respectively, CBT—core body temperature, which was assumed to be equal to the blood temperature, h_t_, h_s1_, h_s2_, h_sa1_, and h_sa2_—the heat transfer coefficients of the skin, sensors 1 and 2, and the interfaces between sensors 1 and 2 and air, respectively, and q_1_ and q_2_—heat flux through sensors 1 and 2, respectively.

**Figure 2 sensors-24-01887-f002:**
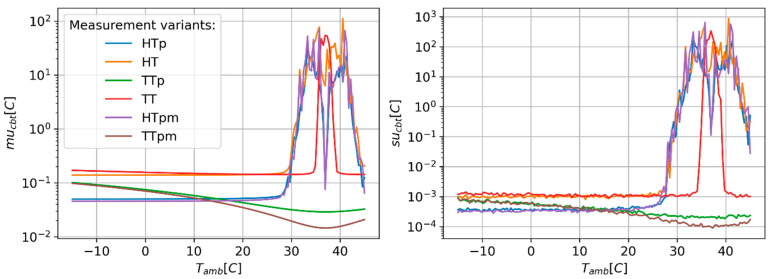
Predicted values of CBT estimation uncertainty (mu_cbt_—(**right**) panel) and corresponding values of standard deviation (su_cbt_—(**left**) panel) of MC simulation repetitions as a function of ambient temperature.

**Figure 3 sensors-24-01887-f003:**
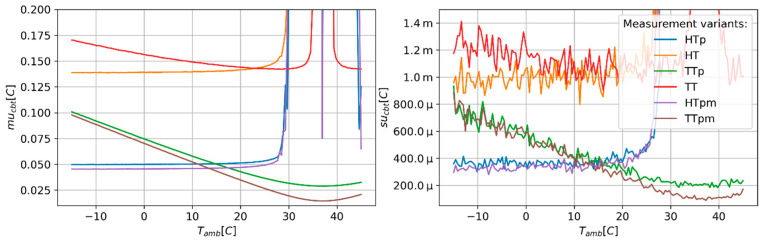
Predicted values of CBT estimation uncertainty (mu_cbt_—(**right**) panel) and corresponding values of standard deviation (su_cbt_—(**left**) panel) of MC simulation repetitions as a function of ambient temperature. Close up on the stable interval of the mu_cbt_ and su_cbt_ value range.

**Figure 4 sensors-24-01887-f004:**
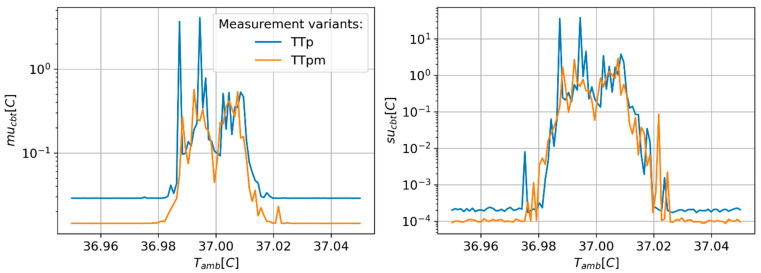
Predicted values of CBT estimation uncertainty (mu_cbt_—(**right**) panel) and corresponding values of standard deviation (su_cbt_—(**left**) panel) of MC simulation repetitions as a function of ambient temperature. Narrowed T_amb_ value range to present the range of unstable behaviour of TTp and TTpm measurement variants.

**Figure 5 sensors-24-01887-f005:**
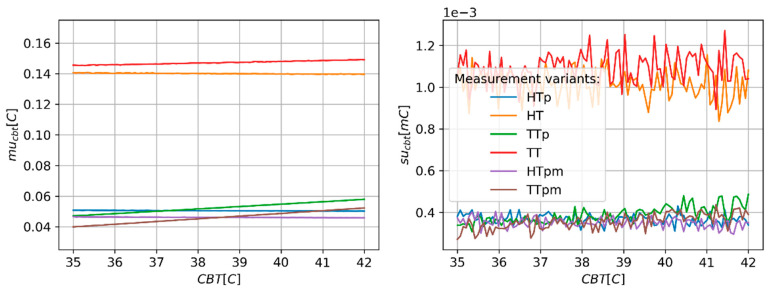
Predicted values of CBT estimation uncertainty (mu_cbt_—(**right**) panel) and corresponding values of standard deviation (su_cbt_—(**left**) panel) of MC simulation repetitions as a function of CBT.

**Figure 6 sensors-24-01887-f006:**
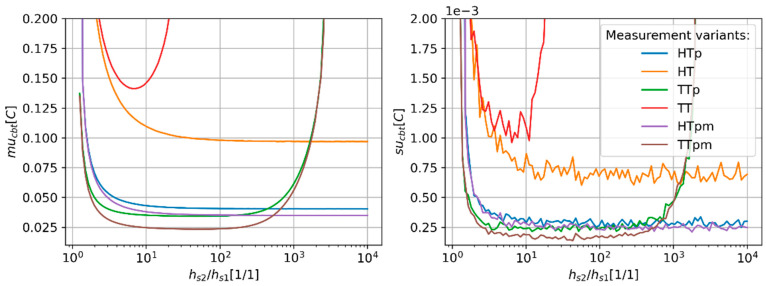
Predicted values of CBT estimation uncertainty (mu_cbt_—(**right**) panel) and corresponding values of standard deviation (su_cbt_—(**left**) panel) of MC simulation repetitions as a function of the heat transfer coefficient of both measurement channels.

**Figure 7 sensors-24-01887-f007:**
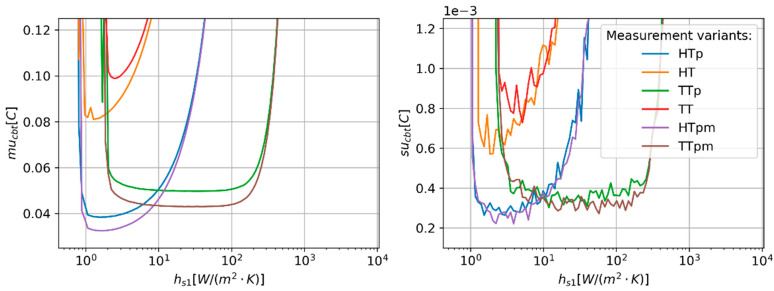
Predicted values of CBT estimation uncertainty (mu_cbt_—(**right**) panel) and corresponding values of standard deviation (su_cbt_—(**left**) panel) of MC simulation repetitions as a function of heat transfer coefficient of sensor 1 (h_s2_ equals 0.5 h_s1_).

**Figure 8 sensors-24-01887-f008:**
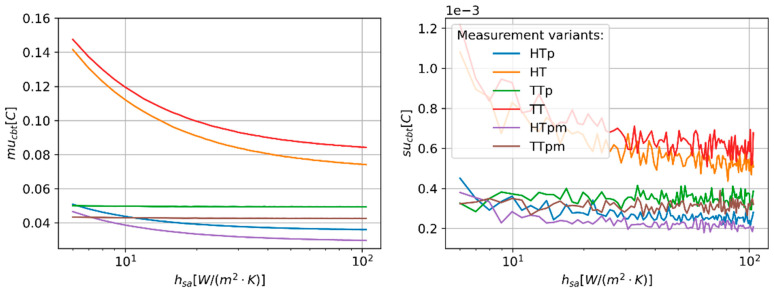
Predicted values of CBT estimation uncertainty (mu_cbt_—(**right**) panel) and corresponding values of standard deviation (su_cbt_—(**left**) panel) of MC simulation repetitions as a function of convection heat transfer coefficient (h_sa_).

**Table 1 sensors-24-01887-t001:** List of all CBT measurement variants used in the study of CBT measurement uncertainty analysis.

Method *	Designation of Independent Measurement Uncertainties (See Section 2.1.2. Monte Carlo Analysis)	Description
HT	u_T_, u_q_	The temperature sensors are not paired.
HTp	u_T_, u_q_, u_ΔT_	The temperature sensors are paired.
HTpm	u_T_, u_q_, u_ΔT_	The temperature sensors are paired. The value of the final estimate is the average value of estimates derived from measurements using Equations (8) and (13).
TT	u_T_	The temperature sensors are not paired.
TTp	u_T_, u_ΔT_	The temperature sensors are paired.
TTpm	u_T_, u_ΔT,_	The temperature sensors are paired. The value of the final estimate is the averaged value of estimates derived from measurements using Equations (9), (14), (15) and (16).

* The CBT sensors come in two types: HT and TT. The HT-type sensors consist of two heat flux and two temperature sensors positioned between the skin and thermoresistive material. The TT-type sensors have four temperature sensors placed at the top and bottom of the thermoresistive material layer of each sub-sensor, as shown in Figure 1.

**Table 2 sensors-24-01887-t002:** List of parameters and their default values used in lumped thermal model of the measurement setup.

Parameter	Unit	Default Value	Description
CBT	[°C]	37	Core body temperature.
T_amb_	[°C]	25	Ambient (air) temperature.
k_s1_, k_s2_	$\left[\frac{W}{m^{2}\cdot K}\right]$	0.15	Thermal conductivity of sensors in channels 1 and 2 as typical material Poly(dimethylsiloxane)(PDMS) was chosen [38].
ds1, ds2	[mm]	15, 30	The thickness of the PDMS layer of sensors in channels 1 and 2, respectively.
k_t_	$\left[\frac{W}{m^{2}\cdot K}\right]$	0.37	Thermal conductivity of the skin [39]
d_t_	[mm]	2.5	Thickness of the skin [40]
h_sa_	$\left[\frac{W}{m^{2}\cdot K}\right]$	6.1	Natural convection coefficient at the surface between sensors and air. Its value was estimated for an upward-oriented circular plate with a radius of 1.5 cm [41,42]. The radius of the plate was chosen based on the description of sensors presented in publications [17,21,35,36,37].
h_s1_, h_s2_	$\left[\frac{W}{m^{2}\cdot K}\right]$	10, 5	Heat transfer coefficient of sensors 1 and 2, respectively. These values were calculated according to the general equation h = k/d.

**Table 3 sensors-24-01887-t003:** Default values of standard uncertainty of different sensors and their elements used in the estimation of combined uncertainty of CBT measurement using dual-heat flux method.

Parameter	Unit	Default Value	Description
u_T_	[°C]	$\pm \frac{0.05}{\sqrt{3}}$	The standard uncertainty of non-paired sensor temperature measurement. The value was estimated based on the datasheet of the LMT70 (Texas Instruments, Dallas, TX, USA) sensor.
u_ΔT_	[°C]	$\pm \sqrt{2}\cdot 0.0003$	The standard uncertainty of paired sensor differential temperature measurement. The value was estimated based on spectral output noise distribution presented in the datasheet of the LMT70 (Texas Instruments, USA) sensor.
u_q_	$\left[\frac{W}{m^{2}}\right]$	±1.3	The standard uncertainty of heat flux measurement. The value was estimated based on the PHFS-01 sensor (Fluxteq, Blacksburg, VA, USA) sensor and AD7713 (Analog Devices, Wilmington, NC, USA) analog-digital converter (ADC) datasheet.
u_h_	$\left[\frac{W}{m^{2}\cdot K}\right]$	±0.12	The standard uncertainty of measurement of the heat transfer coefficients h_s1_ and h_s2_. This value was estimated assuming that the standard uncertainty of sensor thickness equals ±0.1 mm and the relative uncertainty of material conductivity measurement equals ±1% [43].

**Table 4 sensors-24-01887-t004:** List of conducted experiments using Monte Carlo analysis.

Changed Parameter	Unit	Range of Values	Description
T_amb_	[°C]	−15–+45 and 36.95–37.05	The influence of ambient temperature on CBT uncertainty. This test was restated with a narrowed temperature range for TTp and TTpm measurement variants.
CBT	[°C]	35–45	The influence of CBT value on CBT uncertainty.
h_s1_, h_s2_	$\left[\frac{W}{m^{2}\cdot K}\right]$	0.7–6667	The influence of the heat transfer coefficient values of the DHF probe channels on CBT uncertainty. The ratio of the h_s1_ to h_s2_ was constant and equal to 2. The range of the h_s1_ value was selected by changing the thermal conductivity k_1_ and k_2_ in the range from 0.01 W/(m·K) to 100 W/(m·K).
h_s2_/h_s1_	[1/1]	1.25–10,000	The influence of the ratio between heat transfer coefficients of the DHF probe channels on CBT uncertainty. The h_s1_ value was constant as h_s2_ was reduced by increasing the thickness (d_2_) of the second channel.
h_sa_	$\left[\frac{W}{m^{2}\cdot K}\right]$	6–104	The influence of heat transfer coefficient between air and sensor for air flow changing from 0 m/s to 14 m/s on CBT uncertainty. The maximum value of h_sa_ for airflow equal to 14 m/s was estimated using an online [44] calculator assuming that the probe has a size of 2 cm by 2 cm and a temperature equal to 36 °C, and the surrounding air temperature was 25 °C.

**Table 5 sensors-24-01887-t005:** Values of the uncertainty ratio (ru) of a given component to the total uncertainty of CBT measurement using the given measurement variant.

Measurement Variant	ru [%]										
	T_ss1_	T_ss2_	T_sa1_	T_sa2_	q_1_	q_2_	ΔT_ss_	ΔT_s1_	ΔT_s2_	h_s1_	h_s2_
HT	31.832	59.694	-	-	2.947	5.527	-	-	-	-	-
HTp	33.857	-	-	-	22.969	43.074	0.1	-	-	-	-
TT	35.306	62.085	0.141	0.066	-	-	-	-	-	0.48	1.921
TTp	90.048	-	-	-	-	-	0.367	0.001	0.001	1.917	7.667

**Table 6 sensors-24-01887-t006:** Comparison of accuracy of CBT estimation using wearable devices reported in the literature.

Reference	Accuracy [°C]	Device	Type of Experiment
[21]	0.09	DHF	Physical model
[22]	0.3	DHF	Numerical simulation (FEM)
[20]	0.15–0.29	SHF	Clinical experiment
[47]	0.095–0.019	Modified SHF	Numerical (FEM) and limited trials with humans
[31]	0.34	Calera^®^ (greenTEG, Rümlang, Switzerland)	Clinical experiment

## Data Availability

Data are contained within the article.

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
