# Peer review of "Analytical Analysis of Factors Affecting the Accuracy of a Dual-Heat Flux Core Body Temperature Sensor"

_sensors, 2024, doi:10.3390/s24061887_

Round 1

Reviewer 1 Report

Comments and Suggestions for Authors

1. The title "Analytical and numerical analysis", confused.

2. the abstract has the same problem, difficult to understand, and not exactly express the main point of the article.

3. Table 1 is not easy to read, the used sensor is a whole paragraph? the methods should be explained by name and description.

4. have to handle the abbreviation and subscript very carefully, too many parameters and vague abbr. make the article hard to follow.  

5. I still have some reservations about the significance of this research. Can you use experiments to prove the conclusions of these simulations? Because these devices are already in use.    

Comments on the Quality of English Language

1. the abstract need improvement. it is bad if readers have to read repeatedly to catch the idea.

2. too many Abbr.-s are not connected to meaning directly. (reader can not guess the meaning by the Abbr.)

Author Response

We want to express our genuine appreciation for your review of our article and for sharing your valuable feedback. Your insights are instrumental in improving the quality of our work, and we are fully committed to addressing each of your suggestions. Below, we provide a detailed response to your recommendations.

  1. The title "Analytical and numerical analysis", confused.

The intention behind this part of the title was to highlight our utilization of both numerical methods (such as Monte Carlo) and analytical approaches (like first-order Taylor series expansion). However, we understand your concern that the term "analytical" may not adequately convey the distinction we aimed to make. In scientific literature, the term "numerical" often implies an approximate solution obtained through methods of numerical analysis, while "analytical" suggests exact solutions derived through symbolic manipulation. "Analytical" really fails to convey the intended distinction, since both approaches seem analytical, but this terminology seem to be used more frequently.  I acknowledge that this terminology might not effectively communicate the nuances of our work. We have corrected the title to Analytical analysis as this term seems to be wider.

  1. the abstract has the same problem, difficult to understand, and not exactly express the main point of the article.

Thank you for your feedback regarding the abstract of our article. We have taken your comments into serious consideration and have made significant revisions to enhance clarity and ensure that the main points of the article are accurately conveyed. We sincerely hope that the current version of the abstract addresses the concerns raised and provides a clearer and more concise summary of our research findings.

  1. Table 1 is not easy to read, the used sensor is a whole paragraph? the methods should be explained by name and description.

Thank you for your feedback regarding Table 1 in our article. We understand your concern about the readability of the table, particularly the integration of sensor details within a paragraph. We acknowledge that the formatting requirements of the journal may have contributed to this issue, especially considering the prohibition on using row and column separation lines. This constraint has indeed made it challenging to clearly delineate which cells are associated. In response, we have proposed an alternative organization for the table to address this concern. We believe this revised format will enhance readability and clarity, allowing for easier interpretation of the sensor information and method descriptions.

  1. have to handle the abbreviation and subscript very carefully, too many parameters and vague abbr. make the article hard to follow.  

We appreciate your insightful feedback on the utilization of abbreviations and subscripts in our article. Recognizing the complexity arising from the multitude of parameters in our analysis, our aim was to present a thorough and inclusive examination of the subject matter, providing readers with a nuanced understanding of sensor design intricacies. We acknowledge your concern that the extensive use of abbreviations might have compromised clarity and readability. Nonetheless, we believe that the depth of analysis achieved justifies the reader's investment in familiarizing themselves with the abbreviations utilized.

  1. I still have some reservations about the significance of this research. Can you use experiments to prove the conclusions of these simulations? Because these devices are already in use.   

We genuinely appreciate your thoughtful review of our article. Your reservations regarding the significance of our research are duly noted. The primary objective of our work was to initiate a comprehensive exploration into the factors influencing the accuracy of dual heat flux sensors, particularly focusing on CBT measurement precision. While we acknowledge the importance of experimental validation, we opted for a simulation-based approach in this initial phase. This decision was influenced by the considerable time and resources typically required for physical experiments. Our aim was to provide a foundational understanding of the subject matter, offering valuable insights for both practitioners and researchers planning further investigations. We fully agree that experimental validation is crucial, and we are actively pursuing such endeavors in our ongoing research. We are currently in the process of developing experimental setups and sensor models to complement our analytical findings. Additionally, it's worth noting that commercially available sensors, while suitable for practical applications, may not always be optimized for research purposes, given their closed construction. We believe that our article serves as a valuable resource, facilitating more efficient and thoughtful planning of experimental studies in this domain.

  1. the abstract need improvement. it is bad if readers have to read repeatedly to catch the idea.

Thank you for your feedback regarding the abstract of our article. We have taken your comments into serious consideration and have made significant revisions to enhance clarity and ensure that the main points of the article are accurately conveyed. We sincerely hope that the current version of the abstract addresses the concerns raised and provides a clearer and more concise summary of our research findings.

  1. too many Abbr.-s are not connected to meaning directly. (reader can not guess the meaning by the Abbr.)

We are sincerely grateful for your insightful feedback on the use of abbreviations and subscripts in our article. Understanding the complexity inherent in the multitude of parameters analyzed, our goal was to provide a comprehensive examination of sensor design intricacies, facilitating a nuanced understanding for our readers. We acknowledge your concern that the extensive use of abbreviations may have compromised clarity and readability. However, we firmly believe that the depth of analysis justifies the reader's investment in familiarizing themselves with the abbreviations utilized. We have made efforts to ensure that all abbreviations used in the article are closely connected to their meanings while keeping them as concise as possible. Additionally, we have included an abbreviation list in the form of an appendix to assist readers in memorizing the abbreviations. We hope this measure will alleviate any difficulties encountered in understanding the abbreviations and enhance the overall readability of the article.

Reviewer 2 Report

Comments and Suggestions for Authors

The manuscript, which is clear and well-written, shows an analytical and numerical analysis of factors affecting the accuracy of a dual heat flux core body temperature sensor.

The work reported is scientifically sound and convincing. Nevertheless, I would like to suggest the following minor corrections to improve the quality of the manuscript even further.

- some typos have to be removed, and in general, I suggest an extensive language check.

·       the double dot at the end of the caption of Figure 1

·       lines 190 and 243 change “analogically” with “analogously”

·       write the text of the Figures and Tables uniformly in the text in bold or not in bold

·       change in line 370 with “is justified by”.

·       Line 397 change “parried” with “paired”

- Table 1 appears to be difficult to read. I suggest improving the format to understand rows and columns more easily.

-Describe better the sampling procedure of temperature. Explain in detail the temperature range registered with the sensor system.

-Insert a paragraph or table in which you compare your results in terms of accuracy with other works in which different models are applied, if available.

Author Response

We want to express our genuine appreciation for your review of our article and for sharing your valuable feedback. Your insights are instrumental in improving the quality of our work, and we are fully committed to addressing each of your suggestions. Below, we provide a detailed response to your recommendations.

- some typos have to be removed, and in general, I suggest an extensive language check.

  • the double dot at the end of the caption of Figure 1
  • lines 190 and 243 change “analogically” with “analogously”
  • write the text of the Figures and Tables uniformly in the text in bold or not in bold
  • change in line 370 with “is justified by”.
  • Line 397 change “parried” with “paired”

 Thank you for your detailed review and constructive feedback on our article. We have diligently addressed all the issues you pointed out, including the removal of typos and ensuring uniformity in the language throughout the text. To ensure the highest standards of language accuracy, we have enlisted the assistance of the language proofreading office at our university. We are confident that these revisions have significantly enhanced the quality of our article, and we sincerely appreciate your feedback in helping us improve our work.

- Table 1 appears to be difficult to read. I suggest improving the format to understand rows and columns more easily.

Thank you for your feedback regarding Table 1 in our article. We understand your concern about the readability of the table, particularly the integration of sensor details within a paragraph. We acknowledge that the journal's formatting requirements may have contributed to this issue, especially considering the prohibition on using row and column separation lines. This constraint has made it challenging to delineate which cells are associated clearly. In response, we have proposed an alternative organization for the table to address this concern. We believe this revised format will enhance readability and clarity, allowing for easier interpretation of the sensor information and method descriptions.

-Describe better the sampling procedure of temperature. Explain in detail the temperature range registered with the sensor system.

We appreciate your feedback regarding the need for a clearer description of the methods in our article. We apologize for any confusion and are committed to addressing this issue promptly. However, your suggestion is somehow unclear for us thus we kindly request further clarification on the specific areas of concern you have identified i.e. pointing out the positions in the text where this problem occurs. Your guidance will be invaluable in improving the precision and comprehensibility of our article, and we sincerely appreciate your assistance in this matter.

-Insert a paragraph or table in which you compare your results in terms of accuracy with other works in which different models are applied, if available.

Thank you for your valuable suggestion. We have incorporated a new table into our article that compares the accuracy reported by other researchers. This table provides readers with an overview of how our results align with those from existing literature. We believe that this addition provides valuable context for interpreting our findings in relation to previous research efforts.

Reviewer 3 Report

Comments and Suggestions for Authors

The need to measure core body temperature (CBT) is an urgent task both for sports medicine and other medical applications, as well as for everyday life, so the topic of the work seems important. The authors analyze in detail the factors affecting the accuracy of measurements using non-invasive sensors and demonstrate an approach that allows increasing accuracy through proper processing. The manuscript is well structured, however, several minor revisions are required before acceptance for publication. Comments are below:

  1. The structure of Table 1 is confusing because it is not clear which method the comments in the "Used Sensor" and "Description" columns refer to. It takes a long time to understand the information, and there are still doubts about the correct reading. Please think about how to present the information more clearly and understandably for the untrained reader. There are no such perception problems for Tables 2 and 3, for example.
  2. It is not very clear why the scale of measured temperatures starts from -10 C in Figures 2 and 3? The authors measure the core body temperature of a living person, and such values clearly go beyond physiological boundaries.
  3. As the authors themselves write in the conclusion: "The main limitation of the presented study is the lack of empirical validation and the use of a simplified model". Indeed, a purely theoretical analysis, albeit carefully performed, is of significantly less value without validating the created model on real measurements. The reviewer hopes that authors will perform this type of validation in the future.
Comments on the Quality of English Language

 Minor editing of English language required

Author Response

We want to express our genuine appreciation for your review of our article and for sharing your valuable feedback. Your insights are instrumental in improving the quality of our work, and we are fully committed to addressing each of your suggestions. Below, we provide a detailed response to your recommendations.

  1. The structure of Table 1 is confusing because it is not clear which method the comments in the "Used Sensor" and "Description" columns refer to. It takes a long time to understand the information, and there are still doubts about the correct reading. Please think about how to present the information more clearly and understandably for the untrained reader. There are no such perception problems for Tables 2 and 3, for example.

Thank you for your feedback regarding Table 1 in our article. We understand your concern about the readability of the table, particularly the integration of sensor details within a paragraph. We acknowledge that the formatting requirements of the journal may have contributed to this issue, especially considering the prohibition on using row and column separation lines. This constraint has indeed made it challenging to clearly delineate which cells are associated. In response, we have proposed an alternative organization for the table to address this concern. We believe this revised format will enhance readability and clarity, allowing for easier interpretation of the sensor information and method descriptions.

  1. It is not very clear why the scale of measured temperatures starts from -10 C in Figures 2 and 3? The authors measure the core body temperature of a living person, and such values clearly go beyond physiological boundaries.

We appreciate your feedback regarding Figures 2 and 3 in our article. Your concern about the temperature scale starting from -10°C, particularly given that we are measuring core body temperature (CBT), is duly noted. In our experimental setup, we maintained a fixed CBT value of 37°C while varying the ambient temperature (Tamb) from -15°C to 45°C. While this range may appear broader than typical ambient temperatures in populated areas, we believe it holds relevance for potential applications of CBT sensors in extreme environments, such as those encountered by soldiers, firefighters, and workers in frozen food storage facilities. Our aim was to assess any potential limitations in using CBT sensors in such contexts. We trust that this clarification addresses your concerns regarding this issue.

  1. As the authors themselves write in the conclusion: "The main limitation of the presented study is the lack of empirical validation and the use of a simplified model". Indeed, a purely theoretical analysis, albeit carefully performed, is of significantly less value without validating the created model on real measurements. The reviewer hopes that authors will perform this type of validation in the future.

We genuinely appreciate your thoughtful review of our article. Your reservations regarding the significance of our research are duly noted. The primary objective of our work was to initiate a comprehensive exploration into the factors influencing the accuracy of dual heat flux sensors, particularly focusing on CBT measurement precision. While we acknowledge the importance of experimental validation, we opted for a simulation-based approach in this initial phase. This decision was influenced by the considerable time and resources typically required for physical experiments. Our aim was to provide a foundational understanding of the subject matter, offering valuable insights for both practitioners and researchers planning further investigations. We fully agree that experimental validation is crucial, and we are actively pursuing such endeavors in our ongoing research. We are currently in the process of developing experimental setups and sensor models to complement our analytical findings. Additionally, it's worth noting that commercially available sensors, while suitable for practical applications, may not always be optimized for research purposes, given their closed construction. We believe that our article serves as a valuable resource, facilitating more efficient and thoughtful planning of experimental studies in this domain.

Comments on the Quality of English Language

 Minor editing of English language required.

To ensure the highest standards of language accuracy, we have enlisted the assistance of the language proofreading office at our university. We are confident that these revisions have significantly enhanced the quality of our article, and we sincerely appreciate your feedback in helping us improve our work.